# Seizures Related to Influenza in Pediatric Patients: A Comparison with Seizures Associated with Other Respiratory Viral Infections

**DOI:** 10.3390/jcm10143088

**Published:** 2021-07-13

**Authors:** Ji Yoon Han, Seung Beom Han

**Affiliations:** 1Department of Pediatrics, College of Medicine, The Catholic University of Korea, Seoul 06591, Korea; han024@catholic.ac.kr; 2Department of Pediatrics, Daejeon St. Mary’s Hospital, The Catholic University of Korea, Daejeon 34943, Korea; 3The Vaccine Bio Research Institute, College of Medicine, The Catholic University of Korea, Seoul 06591, Korea

**Keywords:** influenza, seizures, respiratory virus, child

## Abstract

Although febrile seizures are the most common neurological complications of influenza, there are few studies comparing seizure characteristics and outcomes between patients with influenza and those with other respiratory virus (RV) infections. Medical records of pediatric patients presenting with seizures accompanied by fever, in whom RV infections were identified, were retrospectively reviewed to compare the characteristics and outcomes of seizures with fever due to influenza (*n* = 97) to those due to other RV infections (*n* = 113). Patients with influenza were older than those with other RV infections (*p* < 0.001), and 22.7% of them were aged ≥5 years. Seizure characteristics of complex febrile seizures were observed more frequently in patients with other RV infections than in those with influenza; however, the frequency of epilepsy was comparable between the two groups. For patients with influenza, children aged <5 years and those aged ≥5 years showed similar seizure characteristics and outcomes. Further neurological evaluations should not be based solely on patient age in children with influenza who experience late-onset seizures at ≥5 years of age. Long-term sequelae should be further investigated in these patients.

## 1. Introduction

Febrile seizure (FS) is defined as a seizure with fever (body temperature over 38 °C), which is not caused by central nervous system (CNS) problems, occurring in children aged between 6 and 60 months without neonatal and unprovoked seizures [1]. Fever in FS patients is commonly caused by respiratory virus (RV) infection rather than severe bacterial infection [2]; therefore, the influenza virus is one of the major pathogenic causes of FS [3]. The influenza virus causes various neurological complications, such as febrile and afebrile seizures, encephalopathy, acute necrotizing encephalopathy, meningitis, encephalitis, myelitis, Reye’s syndrome, and Guillain–Barré syndrome [4,5]. Seizure is the most common among them, and most seizures manifest as FS [4,5]. In our hospital, we encountered seizures accompanied by fever due to influenza virus infection in some older children and adolescents who were beyond the age range of FS [6]. These patients might have seizure characteristics and outcomes that are different from those of patients with FS due to the influenza virus or other RV infections. However, few comparative studies on the characteristics and outcomes of seizures accompanied by fever between pediatric patients infected by the influenza virus and those infected by other RVs have been reported [7]. We conducted a retrospective study of pediatric patients with seizures accompanied by fever in a single university-affiliated hospital, and compared seizure characteristics and outcomes between patients infected by the influenza virus and those infected by other RVs.

## 2. Materials and Methods

### 2.1. Patients and Study Design

Between January 2015 and December 2019, pediatric patients aged <19 years who presented with seizures accompanied by fever to the Department of Emergency Medicine or Pediatrics of Daejeon St. Mary’s Hospital (Daejeon, Republic of Korea) were recruited. Daejeon, with a population of 1,458,463 people (as of March 2021), is one of the seven metropolitan cities in Korea, and consists of a central urban area and surrounding rural areas. Our hospital is a 660-bed, university-affiliated, secondary hospital, and is the fourth largest hospital in Daejeon. Annually, approximately 1800 patients from both urban and rural areas are hospitalized in the Department of Pediatrics. Korea is a monoethnic country; thus, only Koreans were considered for this study. Among them, patients infected by RVs, including the influenza virus, were included in this study. Patients with (1) a history of epilepsy and unprovoked seizures; (2) underlying CNS problems due to perinatal insults, infection, and trauma; and (3) electrolyte or metabolic abnormalities were excluded. Furthermore, patients with underlying neurodevelopmental disorders, including autism spectrum disorders, developmental delays, and genetic disorders, were excluded because of their significant association with seizure disorders/epilepsy [8]. During the study period, each episode of seizure accompanied by fever in the same patient was considered as a separate episode if fever and seizure were absent for ≥7 days and ≥24 h, respectively, between seizure episodes. Clustered seizures which occurred within 24 h were considered a single episode, and the number of seizures within 24 h was investigated.

The medical records of the included patients were retrospectively reviewed to collect their demographics, seizure characteristics and outcomes, and electroencephalography (EEG) and brain magnetic resonance imaging (MRI) findings. The investigated data were compared between patients infected by the influenza virus (influenza group) and those infected by RVs other than the influenza virus (other RVs group). For the influenza group, the investigated data were further compared between patients with FS (<5 years of age) and those with late-onset seizures (≥5 years of age). This study was approved by the Institutional Review Board of Daejeon St. Mary’s Hospital, with a waiver for the requirement to obtain informed consent (approval number: DC21RAS1004).

### 2.2. Definition

Influenza was diagnosed when at least one of two virological tests identified influenza A or B viruses. A commercial rapid influenza diagnostic test (RIDT) kit (Alere BinaxNOW^®^ Influenza A & B Card, Abbott, IL, USA) and multiplex polymerase chain reaction (mPCR) test kit (Anyplex^®^ II RV16 Detection Kit, Seegene Inc., Seoul, Korea) were used to identify RVs in a nasopharyngeal swab sample. In our hospital, mPCR tests for RVs were performed for hospitalized patients presenting with respiratory symptoms or fever without localizing signs, unless the guardian refused testing. During the influenza season, RIDTs were performed prior to mPCR testing, and mPCR tests were performed based on the attending physician’s decision. RV infection, other than influenza, was diagnosed when the mPCR test result was positive for at least one of the tested RVs (adenovirus; parainfluenza virus types 1, 2, 3, and 4; rhinovirus; bocavirus; coronavirus 229E, OC43, and NL63; enterovirus; human metapneumovirus; and respiratory syncytial virus [RSV] groups A and B) excluding the influenza virus. FS was defined as a seizure accompanied by fever (body temperature ≥38 °C) occurring in children aged between 6 and 60 months without other causes of seizures. Seizures that occurred in patients aged >60 months were defined as late-onset seizures.

### 2.3. Statistical Analysis

For comparisons between the influenza and other RVs groups and between the FS and late-onset seizure groups, Kruskal–Wallis and chi-square tests were used for continuous and categorical factors, respectively. Statistical analysis was performed using SPSS 21 software (IBM Corp., Armonk, NY, USA), and statistical significance was set at *p* value < 0.05.

## 3. Results

During the study period, a total of 717 episodes of seizures accompanied by fever were identified in the pediatric patients (Figure 1). Among them, RV tests were performed for 405 (56.5%) episodes: RIDTs for 319, mPCR tests for 86, and both for 69 episodes. The influenza group included 108 seizure episodes where influenza viruses were identified (98 and 10 episodes by RIDTs and mPCR tests, respectively), and the other RVs group included 114 episodes where RVs other than the influenza virus were identified by mPCR tests. Five patients with the influenza virus and other RVs who were concurrently identified by mPCR tests were excluded from the influenza group. Influenza A and B viruses were identified in 68 and 33 episodes, respectively, and both influenza A and B viruses were identified in two episodes. Underlying neurodevelopmental disorders were identified in seven (3.2%) episodes (six [5.8%] in the influenza group vs. one [0.9%] in the other RVs group, *p* = 0.055), and statistical analyses were performed excluding these patients (Figure 1). Eventually, 210 episodes of seizures with fever in 189 patients were included in this study: thirteen and four patients experienced seizures with fever two and three times during the study period, respectively.

### 3.1. Comparison between Influenza and Other Respiratory Viruses Groups

The median age of patients in the influenza group was higher than that of patients in the other RVs group, and significantly more children aged ≥5 years were included in the influenza group than in the other RVs group (*p* = 0.001, Table 1, Figure 2). In patients aged ≥5 years, 75.9% of them were included in the influenza group (Table 1).

Most seizures were generalized tonic-clonic seizures in both groups. Although most seizures occurred once a day and discontinued within 5 min, seizure episodes in the other RVs group were more likely to be longer (*p* = 0.018) and to repeat in a day (*p* < 0.001) than those in the influenza group (Table 1).

EEG and brain MRI scans were performed in 77 and 39 episodes, respectively, and the frequencies of abnormal EEG and brain MRI results were comparable between the influenza and other RVs groups (Table 1). Frequencies of subsequent seizures with fever and epilepsy were determined in 108 (51.4%) patients who were followed up for ≥3 months after the seizure episode: 35 (34.3%) patients for a median of 24 months (range, 5–59) in the influenza group and 73 (64.0%) patients for a median of 19 months (range, 3–71) in the other RVs group. Epilepsy was subsequently diagnosed in 5.6% of the patients, and 42.6% of the patients experienced further seizures with fever without significant differences in the frequencies between the two groups (Table 1). The comparison between the influenza group and other RVs group excluding seven patients aged ≥5 years (patients with FS), showed identical results (Appendix A).

Among the 181 children aged <5 years, 50 (27.6%) experienced complex FS. Complex FS was more frequently identified in patients in the other RVs group than in those in the influenza group (39.6% vs. 10.7%, *p* < 0.001), and subsequent epilepsy was more frequently diagnosed in patients experiencing complex FS than in those with simple FS (15.4% vs. 1.4%, *p* = 0.017).

### 3.2. Subgroup Analysis in the Influenza Group According to the Patient Ages

For the influenza group, two subgroups were compared by the patient ages: FS group and late-onset seizure group (Table 2). Although the seizure duration was longer in the late-onset seizure group than that in the FS group, seizure type and frequency in a day were comparable between the two groups (Table 2). There were no significant differences in the frequencies of abnormal EEG and brain MRI results, further seizures with fever, and subsequent epilepsy between the two groups (Table 2).

## 4. Discussion

In this study, seizure characteristics and outcomes were compared between pediatric patients infected by the influenza virus and those infected by other RVs. Furthermore, patients infected by the influenza virus were compared between patients with FS and those with late-onset seizures. Although there were more patients aged ≥5 years who were infected by the influenza virus than those infected by other RVs, seizure outcomes were not significantly different according to the infecting RV and patient ages.

FS is an age-related episode occurring in children younger than 60 months, and is the most common cause of pediatric neurology or emergency department visits. Most episodes of FS are self-limiting, and the risk of developing epilepsy after FS is low [9]. However, a complex FS (prolonged seizure, focal seizures, and seizure recurrence within 24 h) implies a higher risk of later progression to epilepsy [10]. The prevalence of FS in Europe and the USA was estimated to be around 2–5%, while, in Japan and Korea, the prevalence is 7–8% [11,12,13]. We encountered some pediatric patients with influenza who presented with seizures accompanied by fever beyond 5 years of age [6]. This study was performed to evaluate whether the characteristics and outcomes of seizures accompanied by fever in influenza patients aged ≥5 years were different from those of FS patients due to influenza and other RV infections. Although influenza is regarded as a self-limiting viral illness in previously healthy children, several neurological complications, ranging from mild simple FS to severe necrotizing encephalopathy, can occur [4,5,14,15]. Whether late-onset seizures in patients with influenza could be a prodrome of severe neurological complications was also questioned. In this study, although significantly more children in the influenza group were aged ≥5 years than in the other RVs group, seizure characteristics consistent with complex FS were more frequently observed in the other RVs group than in the influenza group. Moreover, the frequencies of EEG and brain MRI abnormalities, the subsequent recurrence of seizures accompanied by fever, and the development of epilepsy were not significantly different between the two groups. In a subgroup analysis of the influenza group, old age had no significant impact on seizure characteristics or clinical outcomes. Therefore, the decision to perform further neurological evaluations in older children experiencing seizure and fever by influenza virus infection should not be based solely on patient age. Neurological complications were reported in 7.6% of hospitalized pediatric patients with influenza, and 44.4% and 27.8% of them experienced complications other than seizures and received intensive care, respectively [5]. The presence of neurological symptoms and signs suggesting complications other than seizures, such as meningeal signs, altered mentality, and focal neurological deficits, requires further neurological evaluation.

FS is strongly age-dependent; the peak incidence is at 18 months of age, and 85–90% of FS episodes occur before 3 years of age [1,16]. The incidence of seizures accompanied by fever was estimated to be 4–6% before the age of 6 months and 6% after 5 years of age [10,17]. Similar to these results, 6.2% of patients in the other RVs group were aged ≥5 years in this study. However, in the influenza group, 22.7% of patients were aged ≥5 years. This difference in age distribution between the influenza and other RVs groups might be due to the differences in the prevalent age for each RV infection. A previous study reported that RSV and parainfluenza virus infections occurred more frequently in younger children, whereas influenza virus and rhinovirus infections occurred more frequently in older children [18]. The national surveillance results for RV infections operated by the Korea Disease Control and Prevention Agency showed that the influenza virus and rhinovirus were major pathogens in both younger and older children, whereas the proportions of RSV, adenovirus, and enterovirus infections were lower in older children than in younger children [19]. However, considering that high body temperature provoked seizures in immature rodents [20], brain immaturity should be associated with the development of FS in young children aged <5 years with developing brains. The blood–brain barrier is completely established within 6 months of age, but brain maturation has not been completed [21]. Although the sprouting of axons and dendrites is completed at approximately 5 years of age, myelination continues to adolescence [22]. The brain volume reaches 70% of adult brain volume by 12 months of age, 80% of adult brain volume by 24 months of age, and 90% of adult brain volume by 5 years of age [23,24]. Although the different prevalent ages according to the types of RVs explained the different infection rates of various RVs between younger and older children, they could not explain why seizures occurred in older children with maturing brains in the influenza group. Previous studies, as well as this study, have reported that a higher proportion of patients aged ≥5 years were infected by the influenza virus than by other RVs among pediatric patients presenting with seizures accompanied by fever [4]. This may imply that the pathogenesis of seizures accompanied by influenza is different from that of FS due to other RVs. Previous studies reported increased serum and cerebrospinal fluid concentrations of IL-1β, IL-6, IL-10, and TNF-α, as well as increased transcription of their coding genes in patients with influenza-associated encephalopathy and FS [25,26,27]. Cytokine concentrations were correlated with the severity of neurological complications [26,27]. Pro-inflammatory cytokines can directly influence neuronal activity by alternating voltage-gated ion channels and increasing calcium ion permeability of N-methyl-D-aspartate and α-amino-3-hydroxy-5-methyl-4-isoazolepropionic acid receptors, and eventually promote neuronal excitability, seizure generation, and neuronal injury [28,29]. In addition to systemic inflammatory reactions, direct CNS invasion by the influenza virus might be responsible for the pathogenesis of neurological complications [30]. Considering that influenza viral affinity to host receptors and neurotropism seem to differ according to influenza virus subtypes [30], future studies should evaluate pathogenesis and age distributions in patients with neurological complications of influenza separately according to influenza virus subtypes.

In this study, seizure characteristics consistent with complex FS were more frequently observed in the other RVs group than in the influenza group. Since the other RVs group included patients infected by a variety of RVs, the frequency of complex FS in each RV infection should be independently investigated. Although our previous study did not find significant differences in seizure characteristics and outcomes according to the type of infected RVs, including the influenza virus [3], both the previous and present studies included a small number of patients and multiple viruses were concurrently identified by an mPCR test in approximately half of the included patients. Future studies should include a sufficient number of patients and should be performed considering patient age and race, study region, accessibility to health service, and the type of infecting RV.

There are some limitations to this study, including its retrospective design. We analyzed patients from a single hospital, including a relatively small number of patients. Because subtypes of influenza A and B viruses were not determined in our hospital, differences in seizure characteristics and outcomes in the influenza group could not be determined according to influenza virus subtypes. Data on seizure characteristics were omitted in some patients; however, their proportions in the whole subject were marginal. Approximately half of the included patients were followed up for <3 months, and the remaining patients were followed up for a median of 21 months; therefore, long-term sequelae of seizures could not be determined. Although older age was not significantly associated with adverse seizure outcomes in this study, careful long-term follow-up should be planned for older children experiencing seizures accompanied by fever. EEG and brain MRI were performed in a small portion of the included patients; however, they are not usually recommended for patients with simple FS [2]. To overcome these limitations, a prospective multicenter study should be performed for several years to include an appropriate number of patients infected by various subtypes of the influenza virus and various RVs, and to determine long-term sequelae of seizures accompanied by fever.

## 5. Conclusions

In conclusion, pediatric patients infected by the influenza virus tended to experience seizures accompanied by fever at a more advanced age compared to those infected by other RVs, without any significant differences in clinical outcomes. Further neurological investigations should not be performed solely based on patient age for pediatric patients with influenza who experience late-onset seizures at ≥5 years of age. However, more information on long-term sequelae in these patients should be acquired through investigation.

## Figures and Tables

**Figure 1 jcm-10-03088-f001:**
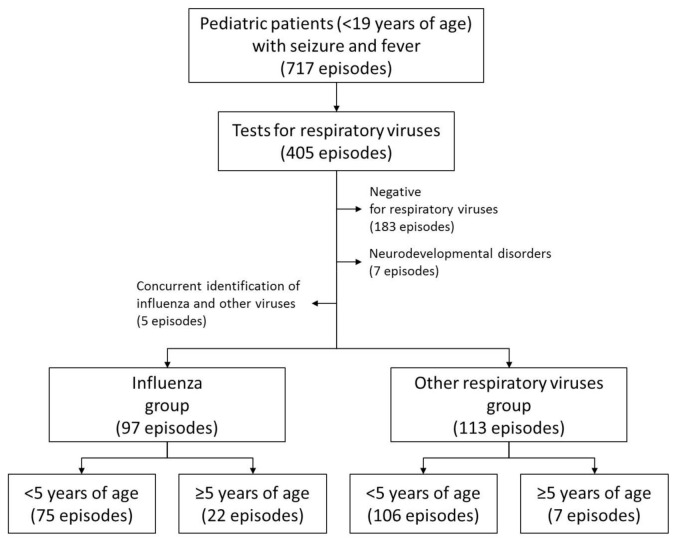
Flow chart for study subjects.

**Figure 2 jcm-10-03088-f002:**
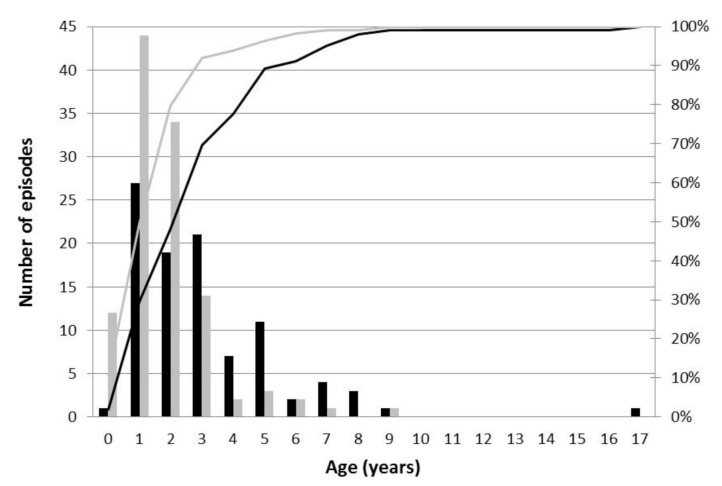
Number of episodes of seizures accompanied by fever along with their accumulated ratios according to the patient ages in the influenza group (black color) and in the other respiratory viruses group (gray color).

**Table 1 jcm-10-03088-t001:** Comparison between the influenza and other respiratory viruses groups.

Factor	Influenza Group (*n* = 97)	Other RVs Group (*n* = 113)	*p*-Value
Median age, years (range)	3 (0–17)	2 (0–9)	<0.001
Age group			0.001
<5 years	75 (77.3)	106 (93.8)	
≥5 years	22 (22.7)	7 (6.2)	
Sex			0.797
Male	61 (62.9)	73 (64.6)	
Female	36 (37.1)	40 (35.4)	
Type of seizures			0.998
Generalized tonic-clonic	79/93 (84.9)	95/112 (84.8)	
Generalized tonic	10/93 (10.8)	12/112 (10.7)	
Others	4/93 (4.3)	5/112 (4.5)	
Seizure duration			0.018
≤5 min	85/94 (90.4)	87/112 (77.7)	
≤15 min	9/94 (9.6)	15/112 (13.4)	
≤30 min	0/94 (0.0)	8/112 (7.1)	
>30 min	0/94 (0.0)	2/112 (1.8)	
Number of seizures within 24 h			<0.001
One	88 (90.7)	76 (67.3)	
Two	7 (7.2)	30 (26.5)	
Three or more	2 (2.1)	7 (6.2)	
Number of previous seizures with fever			0.545
Null	42/96 (43.8)	45/113 (39.8)	
One	23/96 (24.0)	21/113 (18.6)	
Two	10/96 (10.4)	16/113 (14.2)	
Three or more	21/96 (21.9)	31/113 (27.4)	
Time interval between fever and seizures			0.871
<6 h	30/93 (32.3)	42/109 (38.5)	
<12 h	21/93 (22.6)	24/109 (22.0)	
<24 h	21/93 (22.6)	24/109 (22.0)	
<72 h	20/93 (21.5)	18/109 (16.5)	
≥72 h	1/93 (1.1)	1/109 (0.9)	
Abnormal EEG result	2/23 (8.7)	5/54 (9.3)	1.000
Abnormal brain MRI result	0/12 (0.0)	2/27 (7.4)	1.000
Family history of febrile seizures	24 (24.7)	40 (35.4)	0.094
Family history of epilepsy	1 (1.0)	0 (0.0)	0.462
Subsequent seizures with fever	16/35 (45.7)	30/73 (41.1)	0.650
Subsequent diagnosis of epilepsy	1/35 (2.9)	5/73 (6.8)	0.662

RV: respiratory virus; EEG: electroencephalography; MRI: magnetic resonance imaging.

**Table 2 jcm-10-03088-t002:** Comparison between the febrile seizure and late-onset seizure groups in the influenza group.

Factor	Febrile Seizure Group (*n* = 75)	Late-Onset Seizure Group (*n* = 22)	*p*-Value
Sex			0.112
Male	44 (58.7)	17 (77.3)	
Female	31 (41.3)	5 (22.7)	
Type of seizures			0.133
Generalized tonic-clonic	64/72 (88.9)	15/21 (71.4)	
Generalized tonic	6/72 (8.3)	4/21 (19.0)	
Others	2/72 (2.8)	2/21 (9.5)	
Seizure duration			0.024
≤5 min	69/73 (94.5)	16/21 (76.2)	
≤15 min	4/73 (5.5)	5/21 (23.8)	
Number of seizures within 24 h			0.226
One	67 (89.3)	21 (95.5)	
Two	7 (9.3)	0 (0.0)	
Three or more	1 (1.3)	1 (4.5)	
Number of previous seizures with fever			0.242
Null	35/75 (46.7)	7/21 (33.3)	
One	19/75 (25.3)	4/21 (19.0)	
Two	8/75 (10.7)	2/21 (9.5)	
Three or more	13/75 (17.3)	8/21 (38.1)	
Time interval between fever and seizures			0.963
<6 h	24/73 (32.9)	6/20 (30.0)	
<12 h	17/73 (23.3)	4/20 (20.0)	
<24 h	16/73 (21.9)	5/20 (25.0)	
<72 h	15/73 (20.5)	5/20 (25.0)	
≥72 h	1/73 (1.4)	0/20 (0.0)	
Abnormal EEG result	0/10 (0.0)	2/13 (15.4)	0.486
Abnormal brain MRI result	0/2 (0.0)	0/10 (0.0)	NA
Family history of febrile seizures	20 (26.7)	4 (18.2)	0.417
Family history of epilepsy	1 (1.3)	0 (0.0)	1.000
Subsequent seizure with fever	15/30 (50.0)	1/5 (20.0)	0.347
Subsequent diagnosis of epilepsy	0/30 (0.0)	1/5 (20.0)	0.143

EEG: electroencephalography; MRI: magnetic resonance image; NA: not applicable.

## Data Availability

The data presented in this study are available on request from the corresponding author. The data are not publicly available due to national regulations.

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
