# Peer review of "Seizures Related to Influenza in Pediatric Patients: A Comparison with Seizures Associated with Other Respiratory Viral Infections"

_jcm, 2021, doi:10.3390/jcm10143088_

Round 1
Reviewer 1 Report
This study was very interesting to understand the features of fever-related seizures associated with influenza. However, there are some problems.
Materials and Method
The authors should describe the followings.
- The research period, the information research region such as population and race, and characteristics of the hospital where the patients visited.
- How to select patients. The authors should explain which they collected only FS patients or all patients with fever who had seizures. The description is very important to discuss the examination for elderly children with influenza who have seizures.
- The definition of one episode.
- The relation between episodes and patients. The authors should describe how to statistically handle patients who had seizures multiple times.
Results
The expression that the influenza group is older is a misleading expression. The expression that the average age is high is correct. Both modes are 1 year old, and the distribution pattern is basically the same. However, influenza is widespread among older children.
Minor points
In Figure1, the authors should add “episodes” to the numbers.
Reviewer 2 Report
An interesting retrospective observational study examining the pattern of seizures associated with fever seen in children testing positive for influenza vs other respiratory viruses. Important findings are that despite older than expected age for seizures associated with fever in influenza patients, outcomes were generally reassuring.
While the findings are of interest, the authors should be cautious in their conclusions based on the observational study design due to its limitations. While the short term outcomes are reassuring, there is greater interest in long term outcomes which a retrospective study such as this can only provide limited information.
Abstract: page 1 line 25. I would be cautious about the statement that “Further neurological evaluations may not be necessary for older children experiencing seizures with fever due to influenza.” Since influenza can be associated with more serious presentations including encephalitis which can present initially with seizures (Clin Infect Dis. 2017 Aug 15;65(4):653-660. doi: 10.1093/cid/cix412), it is important to not to overstate the findings of this small study. Clinical judgement of the treating physician would still be most important in determining the need for further investigations in older children with influenza presenting with seizures.
Materials and methods. Page 2 line 50.
More details are required regarding the methods. How were cases selected, were there any exclusion criteria, what are the thresholds for doing RV testing within the institution, what outcomes were measured or extracted from patient files e.g investigations performed, imaging etc.
It is unclear why were patients co-infected with influenza and other respiratory viruses automatically analysed with influenza patients?
Results
Page 2 line 77. No details are provided on other co-morbidities within patients besides neuromuscular disorders, eg. neurodevelopmental disorders, epilepsy. Why were “neuromuscular” disorders specifically excluded? Are the authors no other confounders could explain the differences in age distributions observed, differences in length of seizures, and frequency of repeated seizures in an illness, in the two patient groups.
Table 1 and Table 2 pages 4 and 5:
Given denominators are so different for each outcome, it make more sense to highlight percentages and have numbers next to them (n/N). There is no footnote 8 in table 1 regarding what number of patients had information on subsequent diagnosis of epilepsy.
Page 5 Line 111.
The authors should comment if outcomes were different between patients who presented with a pattern of complex vs simple type febrile seizures.
Discussion:
Page 6 line 155.
The authors should be careful to qualify their findings on outcomes by the fact that longer term follow up was quite limited as stated in page 7 line 198 with half of patients having <3 months follow up. It would be difficult to make robust conclusions on risks of epilepsy and neurodevelopmental outcomes based on such limited follow up. Less than half of patients appeared to have information on subsequent seizures or epilepsy.
Round 2
Reviewer 1 Report
Authors well revised the manuscript according to the comments.